# *Citrus reticulata* Leaves Essential Oil as an Antiaging Agent: A Comparative Study between Different Cultivars and Correlation with Their Chemical Compositions

**DOI:** 10.3390/plants11233335

**Published:** 2022-12-01

**Authors:** Nouran M. Fahmy, Sameh S. Elhady, Douha F. Bannan, Rania T. Malatani, Haidy A. Gad

**Affiliations:** 1Department of Pharmacognosy, Faculty of Pharmacy, Ain Shams University, Cairo 11566, Egypt; 2Department of Natural Products, Faculty of Pharmacy, King Abdulaziz University, Jeddah 21589, Saudi Arabia; 3Department of Pharmacy Practice, Faculty of Pharmacy, King Abdulaziz University, Jeddah 21589, Saudi Arabia

**Keywords:** *Citrus reticulata*, GC-MS, antiaging, molecular docking, chemometric analysis, industries development, drug discovery

## Abstract

The mass-based metabolomic approach was implemented using GC-MS coupled with chemometric analysis to discriminate between the essential oil compositions of six cultivars of *Citrus reticulata*. The antiaging capability of the essential oils were investigated through measurement of their ability to inhibit the major enzymes hyaluronidase, collagenase, and amylase involved in aging. GC-MS analysis resulted in the identification of thirty-nine compounds including β-pinene, d-limonene, γ-terpinene, linalool, and dimethyl anthranilate as the main components. Multivariate analysis using principal component analysis (PCA) and hierarchal cluster analysis (HCA) successfully discriminated the cultivars into five main groups. In vitro antiaging activity showed that Kishu mandarin (Km) (2.19 ± 0.10, 465.9 ± 23.7, 0.31 ± 0.01 µg/mL), Cara mandarin (Cm) (3.22 ± 0.14, 592.1 ± 30.1, 0.66 ± 0.03 µg/mL), and Wm (8.43 ± 0.38, 695.2 ± 35.4, 0.79 ± 0.04%) had the highest inhibitory activity against hyaluronidase, collagenase, and amylase, respectively. Molecular docking studies on the major compounds validated the activities of the essential oils and suggested their possible mechanisms of action. Based on our result, certain cultivars of *Citrus reticulata* can be proposed as a promising candidate in antiaging skin care products.

## 1. Introduction

Genus *Citrus* belongs to family Rutaceae (Rue family) and comprises about 17 species and distributed all over the tropical and temperate regions with numerous health benefits [1]. Citrus essential oil (EO) is present in different plant parts such as peels, leaves, and flowers. Terpenes, sesquiterpenes, aldehydes, alcohols, esters, and sterols constitute the major classes of active compounds present in citrus essential oils. They may also be described as mixtures of hydrocarbons, oxygenated compounds and nonvolatile residues. It is worth noting that the essential oil composition of different citrus species is affected by various factors such as the harvest year, due to climate change throughout the years [2], season [3], cultivar [4], the use of different rootstocks [5], as well as the extraction technique [6,7]. Mostly, they are consumed as aroma flavor in the food industry, including alcoholic and nonalcoholic beverages, marmalades, gelatins, sweets, soft drinks, ice creams, dairy products, jams, candies, and cakes [8,9,10]. Moreover, citrus essential oils exhibited significant importance due to their wide range of biological activities such as antimicrobial [11,12,13], antifungal [14], antioxidant [15], antidiabetic [16], antihyperlipidemic [17], anti-inflammatory, anti-allergic [18], anticancer [19], anxiolytic [20], and insecticidal activity [21].

*Citrus reticulata* Blanco, one of the most commercially important species, is commonly known as mandarin fruit [22]. It is native to China, East Asia, and Southeast Asia [23]. The name ‘Mandarin’ was provided to the *C. reticulata* by the Portuguese from its Chinese name Guānhuà, to reflect its country of origin [24]. Different cultivars of *C. reticulata* were traditionally used in folk medicine for the treatment of various ailments such as fever, snakebite, stomachache, edema, cardiac diseases, bronchitis, and asthma [25]. Mandarin oil is well-known for its broad spectrum antibacterial and antifungal actions [26,27,28,29,30]. Anti-proliferative [31], antioxidant [26,32], antidiabetic [30], and schistosomicidal effects [33] were also reported.

On the other hand, nature remains a substantial source for drug discovery for treatment of major ailments as well as in the development of cosmetic products. Currently, the use of natural ingredients in cosmetics is widely spread compared with the synthetic alternative, due to their wide safety margin, potential antioxidant, anti-inflammatory, and skin soothing effects. Interestingly, many of the commercial and medical skin aging creams contain essential oils, herbal extracts, and other natural ingredients which meet the increased demand in the market for herbal-based cosmetics [34]. Moreover, the promising antioxidant activity of various citrus essential oils encouraged us to study the anti-collagenase, anti-hyaluronidase, and anti-elastase capabilities to prove their antiaging potential in an attempt incorporate it safely as a natural ingredient in management of age-related skin problem [35,36,37]. Additionally, the essential oil composition of different *C. reticulata* cultivars showed a significant difference in their chemical constituents [38,39]; thus, a comparative metabolic profiling of poorly studied cultivars is needed for the selection of a good quality cultivar that can be used in improving and upgrading of essential oil composition. The present study targeted a comparative metabolic profiling of the essential oil composition of six cultivars of *Citrus reticulata* leaves cultivated in Egypt using GC-MS and the study of their potential antiaging activity aiming to discover a natural ingredient that can be incorporated safely in antiaging skin care products.

## 2. Results and Discussion

### 2.1. Extraction and Distillation of the Essential Oils

Hydrodistillation of *C. reticulata* fresh leaves cultivars yielded a pale-yellow oil. The yield was expressed as the weight of the oil per 500 g fresh leaves and varied from 0.078 to 0.232% w/w (Table 1).

### 2.2. Metabolic Profile of the Essential Oils

The chemical composition of the essential oils obtained from six cultivars of *C. reticulata* leaves were analyzed using GC-MS. The chemical compounds identified, Kovats indices, and percentages (average of three replicates for each sample) for each cultivar are displayed in Table 2. Monoterpenes hydrocarbons were predominant in all cultivars (36.38–86.45%), oxygenated monoterpenes and sesquiterpenes were present at a lower percentage, other classes of compounds were reported in a high percentage represented chiefly by dimethyl anthranilate in Am (56.51 ± 1.06%), Bm (58.77 ± 2.04%), and Wm (49.06 ± 1.94%) cultivars.

Generally, thirty-nine compounds were tentatively identified including β-pinene, d-limonene, γ-terpinene, linalool, and dimethyl anthranilate as the main components. A great variability in the percentage of the chemical composition among different cultivars was observed. For example, β-pinene content was dominant in Sm cultivar (34.24 ± 3.50%) followed by Km cultivar (19.62 ± 0.37%), while in Wm, Am, and Bm it constitutes only 3.53 ± 0.68%, 2.48 ± 0.11%, and 1.80 ± 0.38% of the essential oil content, respectively. The highest d-limonene content was found in Wm cultivar (12.49 ± 0.21%) while it was totally absent in Sm cultivar. γ-Terpinene was present in all cultivars with amount varying from 48.56 ± 1.01% in Cm cultivar to 6.19 ± 0.31% in Sm cultivar. The highest amount of linalool was present in Km (22.2 ± 0.43%) and represented its major component, contrarily it was absent in Am and Bm. Additionally, certain compounds were present exclusively in certain cultivars in an appreciable amount, for example, m-mentha-6,8-diene was identified only in Sm and accounted for 4.12 ± 0.18% of its essential oil content, while β-thujene and thymol uniquely found in Wm, displayed 3.34 ± 0.10% and 10.98 ± 0.27% of its essential oil content, respectively. 

Previous studies were concerned with the chemical composition of *C. reticulata* leaf essential oils. Lota et al. (2000) reported the variation in the content of γ-terpinene (0.2–61.3%), dimethyl anthranilate (trs–58%), sabinene (0.2–59.4%), linalool (0.2–54.3%), limonene (1.5–44.3%), *p*-cymene (tr–20.4%), β-ocimene (0.6–13.7%), β-pinene (0.1–10.7%), and terpinen-4-ol (0.1–10.6%) among forty-one mandarin cultivars [39]. Moreover, the chemical composition of the essential oils obtained from the leaves of six *C. reticulata* cultivars grown in Nigeria showed a discrepancy in the content of sabinene (1.4–39.7%), γ-terpinene (0.7–32.8%), *p*-cymene (0.3–27.4%), 3-carene (tr–11.6%), and β-ocimene (3.2–19.7%) [24]. These reports validate the difference in the essential oil content between the investigated mandarin cultivars as stated in our study. Another study performed by Karioti et al. (2007) on Nigerian *C. reticulata* displayed a predominance of γ-terpinene (53.0%) and linalool (16.1%) in the leaf essential oil [40], this amount of γ-terpinene is comparable to that found in Cm cultivar, while linalool percentage was near that found in Sm and Km cultivars.

In agreement with our results, Fleisher et al. (1990) reported dimethyl anthranilate as the major constituent of balady mandarin (Bm) which is accredited to 58.77 ± 2.04% of Bm oil content in our study [41]. Previous work on *C. reticulata* grown in Egypt, reported dimethyl anthranilate (65.3%) as a key component of the leaf essential oil followed by γ-terpinene (19.8%) and limonene (4.5%) [29]. These data are in line with the result of Am, Bm, and Wm essential oils. On the other hand, C. Blanco et al. (1995) reported linalool (52.66%), limonene (8.32%), and trans-β-ocimene (7.87%) as the major constituents of Colombian *C. reticulata* leaf oil, these components were found in our studied cultivar ranging from nil to 22.20% for linalool, from nil to 12.49% for limonene, and from 0.06 to 0.31% for trans-β-ocimene [42], which reveal that a difference in the geographical distribution between oils obtained from the same species marks a variation in the essential oil composition.

### 2.3. Chemometric Analysis Based on GC-MS Analysis

Due to the complexity and high dimensionality of GC-MS-based data comprising both qualitative and quantitative variances among different citrus cultivars, multivariate analysis was applied using principal component analysis (PCA) and hierarchal cluster analysis (HCA) to discriminate between closely related cultivars, as well as to detect any significant relationship between them [43]. A matrix of the total number of samples and their replicates (18 samples) multiplied by 39 variables (GC/MS peak area %) was constructed in MS Excel^®^, then subjected to multivariate analysis (PCA and HCA). Owing to the large number of variables, PCA was applied first to reduce the dimensionality of the multiple data set in addition to removing the redundancy in the variables, utilizing raw data (Peak area % for each compound as in Table 2). Figure 1a,b represents PCA score and loading plots based on GC-MS metabolic profiles of different citrus cultivars, respectively.

PCA score plot (Figure 1a) explained about 96% of the variation in the dataset by the first two PCs, where PC1 accounting for 76% and PC2 for 20% of the variance. Different *C*. *reticulata* cultivars were assembled into four main groups on three different quadrants. Cultivars Cm, Sm, and Km were positioned on negative PC1, where Cm was positioned on the upper left quadrant. However, Sm and Km were placed on the lower one, well-discriminated from each other. Through clear investigation of the PCA score plot, it was observed that cultivars Am, Bm, and Wm were located on positive PC1, closely related to each other, where Am and Bm were superimposed on each other. The loading plot (Figure 1b) displayed the main discriminating markers responsible for PCA score plot pattern. Γ-Terpinene, linalool, and β-pinene were the key markers accountable for the segregation of Cm, Km, and Sm, respectively. However, dimethyl anthranilate was the compound responsible for the closeness of Am, Bm, and Wm cultivars.

Additionally, HCA was applied as an unsupervised pattern recognition method to confirm results obtained by PCA. The dendrograms displayed in Figure 1c, revealed segregation of different citrus cultivars into five main clusters. Cluster I, II, and III displayed Km, Sm, and Cm, respectively. HCA dendrograms revealed the closeness of Wm (Cluster IV) to Am and Bm that were grouped together in cluster V. HCA results endorsed that of PCA.

In an attempt to explore the ability of multivariate analysis to discriminate between closely related cultivars (Am, Bm, and Wm), PCA was applied on the GC-MS metabolic profiles of these three cultivars solely. Figure 2a,b displayed PCA score plot and loading plot, respectively. PCA score plot (Figure 2a) explained about 98% of the variation in the dataset by the first two PCs, where PC1 accounting for 95% and PC2 for 3% of the variance. Am, Bm, and Wm were completely segregated from each other in three different quadrants, where Am and Bm were positioned on PC1 on the right side of the plot confirming that they are closely related to each other in comparison with Wm that was located on negative PC1. Upon examination of the loading plot (Figure 2b), it was observed that dimethyl anthranilate was the major marker responsible for discriminating Am from other cultivars (Bm and Wm). However, γ-terpinene and β-caryophyllene were the main distinctive markers accountable for Bm cultivar segregation. Nevertheless, no distinctive marker was recognized in the loading plot for the separation of Wm. From this study, it was concluded that complete metabolic profile is mandatory for discrimination between closely related cultivars. For example, dimethyl anthranilate was the major component in the three cultivars (Am, Bm, and Wm); however, by applying multivariate analysis, it was observed that this marker could only discriminate Am from other cultivars. Similarly, regarding γ-terpinene is present approximately in the same percentage in the three cultivars; however, it discriminated Bm cultivar from other cultivars.

### 2.4. In Vitro Antiaging Activity

The antiaging capability of *C. reticulata* leaf cultivars essential oil was assessed via measuring their hyaluronidase, collagenase, and elastase enzyme inhibitory activity (Figure 3). Hyaluronic acid is one of the major building blocks of soft connective tissues that have a crucial role in maintaining the elasticity and moisture content of the skin, thus reducing wrinkles [44]. Hyaluronidase enzyme converts hyaluronic acid to small oligosaccharide moieties, consequently inhibitors of hyaluronidase are useful in hydrating the skin and delaying the aging process [45,46]. Among the tested cultivars Km and Cm were the most active with an IC_50_ value of 2.19 ± 0.10 and 3.22 ± 0.14 µg/mL, respectively, followed by Wm (IC_50_ = 8.43 ± 0.38 µg/mL) compared with the standard 6-*O*-palmitoyl-l-ascorbic acid exhibiting an IC_50_ of 1.56 ± 0.07 µg/mL. It is worth noting that Km cultivar showed no significant difference from the standard.

Collagen is considered the most abundant protein in the body and plays a crucial role in the structural support, elasticity, strength, and flexibility of the skin connective tissue. Collagenase an enzyme in the matrix metalloproteinase responsible for degrading collagen and thus inhibition of collagenase affects the elasticity of the skin and delays the wrinkling process. Km showed the highest inhibitory action on collagenase with an IC_50_ value of 465.9 ± 23.7 with no significant difference from the standard phenathroline used (IC_50_ = 423.2 ± 21.5 µg/mL). Whereas, elastin protein found in the connective tissue and catalyzed by elastase enzyme is the chief component of elastic fibers and preserves skin elasticity. Elastin degradation is accelerated by age and UV radiation due to the increase in the elastin action, resulting in skin aging [47]. In vitro elastase inhibitory activity revealed that Km, Cm, and Wm were the most active cultivars with IC_50_ of 0.31 ± 0.01, 0.66 ± 0.03, and 0.79 ± 0.04 µg/mL, respectively. However, Km only showed no significant difference from the standard FK 706 used with IC_50_ of 0.15 ± 0.01 µg/mL.

It is obvious from our results that Cm, Km, and Wm exert a promising inhibitory action against tested enzymes. These results may be explained by the difference in the chemical composition of Km and Cm and their segregation from other cultivars as displayed in the PCA score plot and HCA. The pronounced activity of Km might be accredited to the thymol (10.98 ± 0.27%) content uniquely found in this cultivar and linalool (22.20 ± 0.43%) present in the highest percentage in Km. The high γ-terpinene content found in Cm (48.56 ± 1.01 µg/mL) compared with other cultivars might be the reason beyond its enzyme inhibitory activity. Regarding Wm activity, we could not attribute its antiaging activity to the dimethyl anthranilate content only (49.06 ± 1.94 µg/mL), but also to the synergistic effect of the whole essential oil component, as Am and Bm cultivars showing nearly similar dimethyl anthranilate content displayed lower inhibitory activity on the studied enzymes.

### 2.5. In Silico Molecular Docking Studies on the Target Enzymes

The promising inhibitory activity of certain *C. reticulata* cultivars essential oil against hyaluronidase, collagenase, and elastase encouraged us to conduct a docking study of the major identified compounds with the enzymes of interest. This study aimed to discover the potential binding modes in which the essential oils exert its inhibitory action. The major compounds identified were docked into the 3D coordinates of hyaluronidase, collagenase, and elastase using the following PDB IDs: 1fcv, 456c, and 6qeo, respectively. The applied docking parameters were validated by re-docking each co-crystalized ligand into its corresponding active site. The calculated RMSD values between the docked pose and the co-crystalized pose were 0.63, 0.72, and 0.99 Å for hyaluronidase, collagenase, and elastase, respectively, ensuring the validity of the docking protocol. The re-docking of each co-crystalized ligand resulted in docking scores of −8.1, −9.6, and −10.4 Kcal/mole for hyaluronidase, collagenase, and elastase, respectively. The docking of the major compounds to the three enzymes resulted in good acceptable scores, comparable to those of the reference compounds. Table 3 summarizes the docking scores of essential oils major compounds against the three potential target enzymes and their corresponding docking scores.

Interestingly, thymol, dimethyl anthranilate, and linalool and were the most active compounds on the three targets, achieving respective scores of −9.8, −8.2, and −8.1 Kcal/mole with hyaluronidase, −10.2, −8.6, and −7.9 Kcal/mole with collagenase, and −12.3, −8.0, and −9.1 Kcal/mole with elastase, respectively. Inspecting Figure 4, thymol interacted through hydrogen bonds with Asp111, Glu113, Tyr184, Tyr227, and Gln271 and hydrophobic interaction with Trp301, dimethyl anthranilate was found to interact with hyaluronidase through hydrogen bonds with Asp111 and Tyr227 and hydrophobic interactions with Trp267 and Trp301. Similarly, linalool interacted by hydrogen bonds with Asp111, Glu113, Tyr227, and Gln271 and hydrophobic interaction Trp301. As depicted by Figure 5 the top three compounds strongly interacted with the collagenase enzyme in which, thymol forms hydrogen bonds with Gly237, Ala238, and Phe241, besides hydrophobic interaction with Leu239 and Thr245, dimethyl anthranilate formed hydrogen bonds with Ala238, Leu239, Ile243, Tyr244, and Thr245 in addition to hydrophobic interaction with Thr245, while linalool formed six hydrogen bond interactions with Ala238, Phe241, Ile243, Tyr244, and Thr245. Moreover, Figure 6 shows that thymol interacted with Thr41, Cys42, and Cys191 through four hydrogen bond interactions, in addition to one hydrophobic interaction with His57, dimethyl anthranilate engaged in six hydrogen bond interactions with Cys42, His57, and Cys58 found in the active site of the elastase, while linalool interacted through hydrogen bond interactions with Thr41, Cys42, Gln192 and Ser195, in addition to hydrophobic interaction with His57. In conclusion, the three compounds namely, thymol, dimethyl anthranilate, and linalool had the best ability to strongly interact with the three enzymes hyaluronidase, collagenase, and elastase, achieving acceptable docking scores that sometimes exceeded those of the reference compounds. These acceptable scores were achieved through the establishment of many hydrophobic and hydrogen bond interactions. It is worth noting that thymol having the best docking scores in all tested enzyme was identified only in Km, this explains why it exhibited the highest in vitro enzyme inhibition action. Thus, the observed strong binding interactions validated the activities of the essential oils and suggested their possible mechanisms of action.

## 3. Materials and Methods

### 3.1. Plant Material

Leaves were randomly collected from different trees for each cultivar (6 cultivars of *Citrus reticulata*), cultivated under the same climatic conditions and during the fruit ripening stage from Citrus Department botanical garden (geographical coordinates: 30.020111741763642, 31.206797515343563), Horticulture Research Institute, Agriculture Research Center, Giza, Egypt (Table 1). All leaves were collected at the same phenological stage (spring growth cycle) in March 2021 (10 days after flowering, 10th week of the year). Plant materials were botanically identified by Prof. Gamal Farag Abdel Rahman, Head of Citrus Department. Voucher specimens of all collected samples were kept at the Pharmacognosy Department, Faculty of Pharmacy, Ain Shams University with codes (PHG-P-CR-391 to PHG-P-CR-396). 

The type of the soil was light clay soil, (the temperature was about 25 °C with slight rain). The trees were 20 years old, where approximately 5 to 7 trees were used for leaves sampling for each cultivar. The distance of plants between the lines were approximately 1.5 m and 1 m on the line. Phosphate and organic fertilizers and agricultural sulfur were used. Phosphate fertilizers were added in the form of mono-superphosphate at a rate of 30 kg per feddan, during the months of December and January mixed with fully decomposed municipal fertilizers (15–20 m^3^ per feddan) and 100 kg of agricultural sulfur until it decomposed before the spring season. Trees were irrigated by immersion every 30 to 45 days.

### 3.2. Hydro-Distillation of the Essential Oil and GC-MS Analysis

Entire fresh leaves (500 g) were submitted to hydro-distillation within 2 days from collection for 3 h using a Clevenger type apparatus. The oils obtained were recovered, weighed, dried over anhydrous sodium sulfate, and stored in amber and air-tight sealed vials at −20 °C until use. The yield (expressed in % (w/w)) was calculated based on the initial plant weight. The essential oils were analyzed one week after hydro distillation on GC-MS (Shimadzu GCMS-QP 2010, Koyoto, Japan) equipped with Rtx-5MS capillary column (30 m length × 0.25 mm i.d. × 0.25 µm film thickness) (Restek, Bellefonte, PA, USA). The oven temperature was kept at 45 °C for 2 min (isothermal) and programmed to 300 °C at 5 °C/min and kept constant at 300 °C for 10 min (isothermal); injector temperature was 250 °C. Helium was used as a carrier gas with the constant flow rate set at 1.41 mL/min. Diluted samples (1% v/v) were injected with a split ratio 15:1, and the injected volume was 1 μL. The MS operating parameters were as follows: interface temperature: 280 °C; ion source temperature: 200 °C; EI mode: 70 eV; scan range: 35–500 amu. Each sample was analyzed in triplicate [48]. Different compounds of the essential oils were comprehended with the aid of the NIST 05 database (NIST Mass Spectral Database, PC-Version 5.0, 2005, National Institute of Standardization and Technology, Gaithersburg, MD, USA). The Automated Mass Spectral Deconvolution and Identification System (AMDIS 2.64, NIST Gaithersburg, MD, USA) deconvoluted the measured mass spectra. The spectra of individual components were transferred to the NIST Mass Spectral Search Program MS Search 2.0 where they were matched against reference compounds of the NIST Mass Spectral Library 2005.

### 3.3. Multivariate Analysis

The data obtained from GC-MS were subjected to chemometric analysis, Principal Component Analysis (PCA) was applied as an initiative step in data investigation to afford an overview of all cultivar variability and to identify markers responsible for this variation. Hierarchal cluster analysis (HCA) was then utilized to allow clustering of different cultivars. The clustering patterns were built by applying the complete linkage way. This exhibition is more effective when the distance between samples (points) is computed by the Euclidean method [49]. PCA and HCA were accomplished using CAMO′s Unscrambler^®^ X 10.4 software (Computer-Aided Modeling, AS, Oslo, Norway).

### 3.4. In Vitro Antiaging Activity

#### 3.4.1. Hyaluronidase Inhibition Assay

The assay was performed using hyaluronidase inhibitor screening assay kit QuantiChrom^TM^ (BioAssay system, CA, USA) following the manufacturer’s protocol. The assay was based on a turbidimetric reaction by measuring the amount of hyaluronic acid hydrolyzed. In brief, hyaluronidase from bovine testes (Sigma-Aldrich) was prepared freshly in 0.1 M acetate buffer. Serial dilutions of the essential oils were performed using DMSO. Forty microliters of hyaluronidase were transferred in each well of a 96-well plate then 20 µL of tested essential oil was added. The plate was incubated for 15 min at room temperature, then 10 µL substrate and 35 µL buffer were added to the plate, mixed, and incubated for 20 min at room temperature, the decrease in turbidity was measured spectrophotometry at 600 nm. 6-*O*-Palmitoyl-l-Ascorbic Acid (Sigma-Aldrich) was used as a standard hyaluronidase inhibitor.

#### 3.4.2. Collagenase Inhibition Assay

The fluorometric collagenase inhibitor screening kit (Biovision, Catalog # K833-100, CA, USA) was used in the assay following the kit protocol. The kit utilizes Self-Quenched BODIPY conjugate of Type-B gelatin as a fluorogenic substrate to monitor the activity of collagenase. Serial dilutions of the essential oil and the standard (1,10)-Phenanthroline were mixed with 5 µL diluted collagenase and 44 µL collagenase buffer. Fluorescence was measured at 490/520 nm in a kinetic mode at 37 °C for 30–60 min.

#### 3.4.3. Elastase Inhibition Assay

Elastase inhibitory activity was measured following the kit protocol (Molecular Probes’, Catalog # E-12056 EnzChek^®^, Leiden, Netherlands). In brief, fifty microliters of the reaction buffer (pH 8 Tris-HCL buffer) were added into each well together with 50 µL of 100 µg/mL DQ^TM^ elastin substrate working solution to provide a final substrate concentration of 25 µg/mL. Serial dilutions of the oils were added, mixed, and incubated for 30 min at room temperature. Fluorescence intensity was measured using a fluorescence microplate reader equipped with standard fluorescein filters. FK706 was used as a standard elastase inhibitor [50,51].

### 3.5. Statistical Analysis

All experiments were carried out three times in triplicate. Data are expressed as mean ± standard deviation. The IC_50_ values were calculated, and the results were presented using GraphPad Prism^®^ software (Version 7, graph-Pad software Inc., San Diego, CA, USA).

### 3.6. In Silico Molecular Docking Study

The docking studies were conducted using Molecular Operating Environment (MOE 2019.02) Software [52,53]. The X-ray crystal structures of hyaluronidase, collagenase, and elastase were downloaded from the protein data bank using the following PDB IDs: 1fcv, 456c and 6qeo, respectively. Hydrogens and charges of the receptors were optimized using AMBER10: EHT embedded in MOE software. The binding site of the three enzymes was constructed where the corresponding co-crystalized ligand is bound. Major compounds identified in the essential oils were sketched using the 2D builder of MOE2019 and converted to 3D structures using the same software. Compounds were docked into the EGFR binding domain using triangular matcher and London dg as a placement and scoring methods, respectively. At last, 2D and 3D interaction diagrams were generated by MOE to analyze the docking results.

## 4. Conclusions

Metabolic profiling of the essential oils of *C. reticulata* leaf cultivars resulted in the identification of thirty-nine compounds including β-pinene, d-limonene, γ-terpinene, linalool, and dimethyl anthranilate as the main components. Qualitative and quantitative variabilities among the chemical composition of different cultivars was observed using PCA and HCA, which indicate that a complete metabolic profile is mandatory for discrimination between closely related cultivars. Cm, Km, and Wm exerted a promising inhibitory action against tested aging enzymes. In silico studies on the major compounds confirmed the activities of the essential oils and suggested their possible mechanisms of action. From our study we can conclude that certain cultivars of *Citrus reticulata* can be proposed as a promising candidate for incorporation in antiaging skin care products.

## Figures and Tables

**Figure 1 plants-11-03335-f001:**
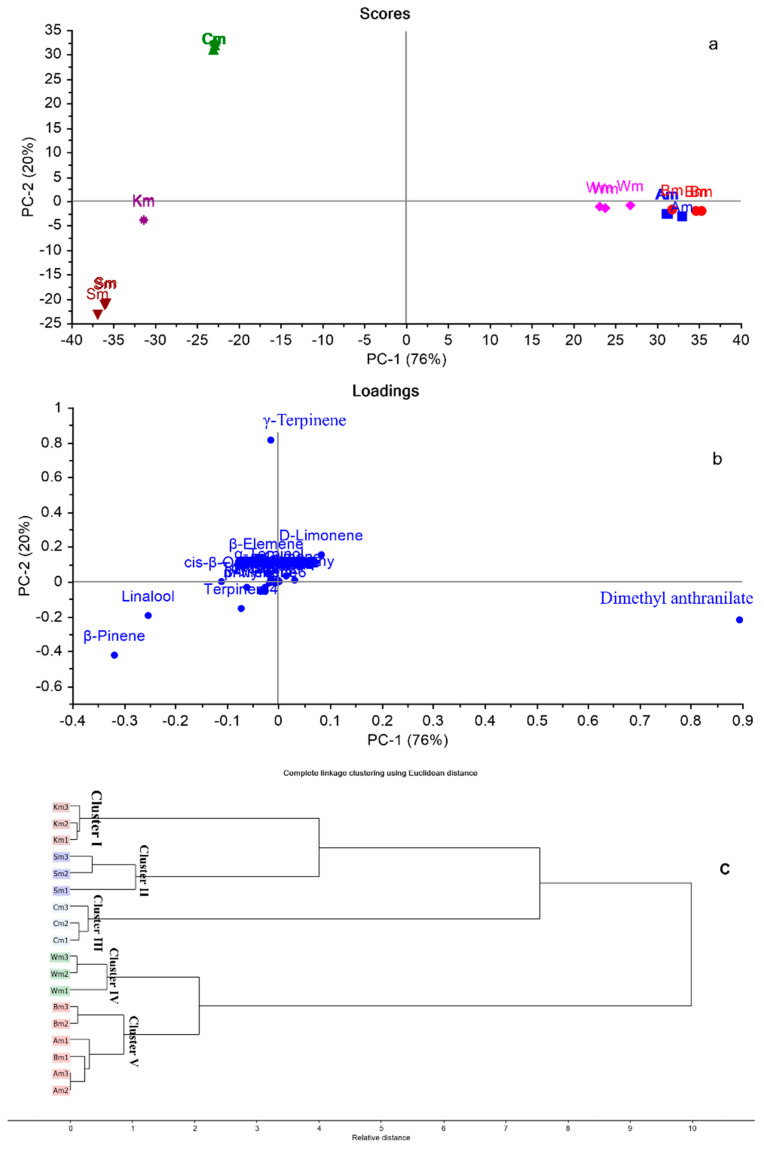
PCA score plot (**a**), loading plot (**b**), HCA dendogram (**c**) based on GC-MS metabolic profiles of different citrus cultivars based on the identification of volatile compounds displayed in Table 2. Refer to Table 1 for cultivar abbreviations.

**Figure 2 plants-11-03335-f002:**
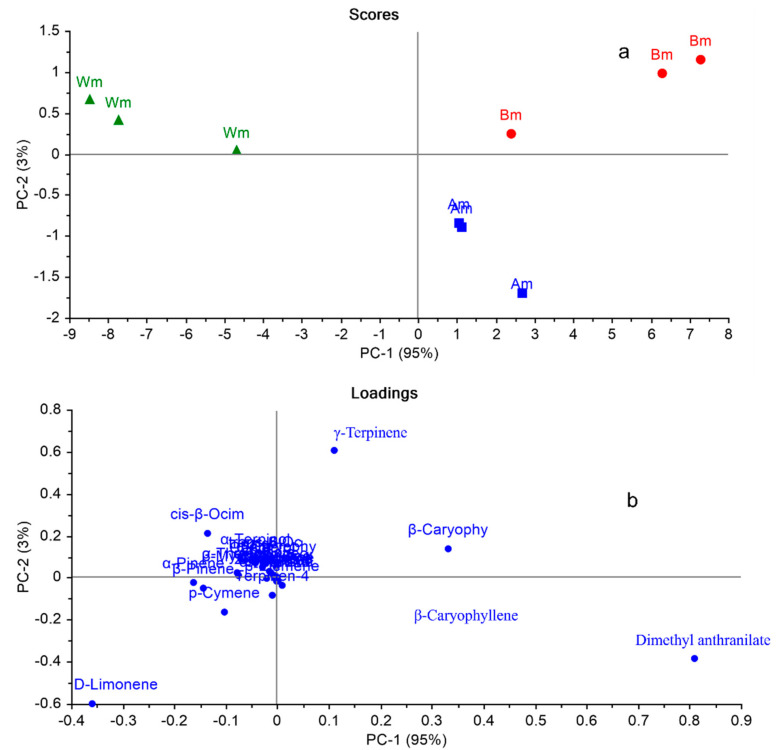
PCA score plot (**a**), loading plot (**b**) based on GC-MS metabolic profiles of Am, Bm, and Wm cultivars based on the identification of volatile compounds displayed in Table 2. Refer to Table 1 for cultivar abbreviations.

**Figure 3 plants-11-03335-f003:**
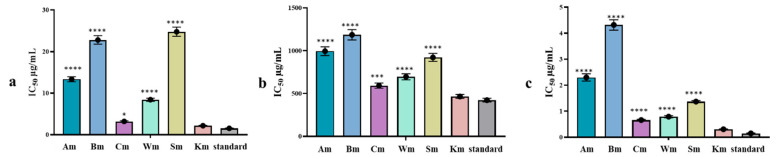
(**a**) Hyaluronidase, (**b**) collagenase, and (**c**) elastase inhibitory activity of *C. reticulata* leaf essential oil cultivar. The results are expressed as the mean ± SD, n = 3. Asterisks indicate significant differences from the standard drug (*, ***, ****, *p* < 0.05, *p* < 0.0009, *p* < 0.0001).

**Figure 4 plants-11-03335-f004:**
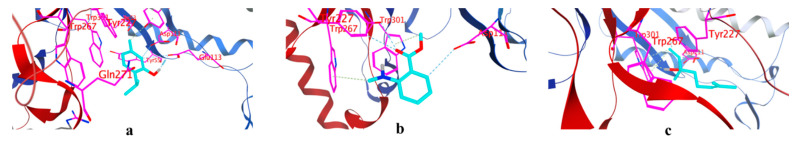
Binding modes of thymol (**a**), dimethyl anthranilate (**b**), and linalool (**c**) with hyaluronidase.

**Figure 5 plants-11-03335-f005:**
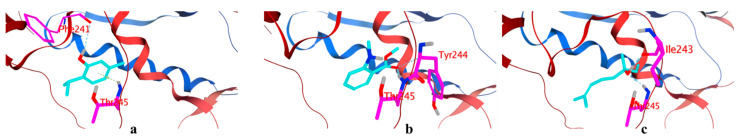
Binding modes of thymol (**a**), dimethyl anthranilate (**b**), and linalool (**c**) with collagenase.

**Figure 6 plants-11-03335-f006:**
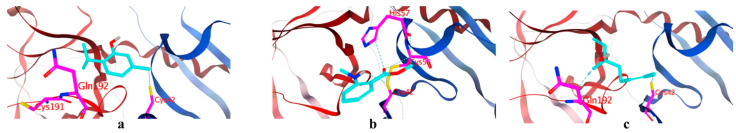
Binding modes of thymol (**a**), dimethyl anthranilate (**b**), and linalool (**c**) with elastase.

**Table 1 plants-11-03335-t001:** *Citrus reticulata* cultivars, codes, and yield % (w/w).

*Citrus reticulata* Cultivar	Cultivar Code	Yield % (w/w)
Avana Apriena mandarin	Am	0.232
Balady mandarin	Bm	0.220
Cara mandarin	Cm	0.124
Willow leaf mandarin	Wm	0.078
Sunburst mandarin	Sm	0.130
Kishu mandarin	Km	0.151

**Table 2 plants-11-03335-t002:** Metabolic profiles of *Citrus reticulata* leaves cultivar EO by GC-MS.

No.	R_t_	Compound Name	KI_exp_ ^a^	KI_rep_ ^b^	Content%						Molecular Formula
					Am	Bm	Cm	Wm	Sm	Km	
1.	7.19	α-Thujene	911	911	0.97 ± 0.05	0.67 ± 0.17	1.37 ± 0.03	1.55 ± 0.63	0.73 ± 0.12	1.08 ± 0.07	C_10_H_16_
2.	7.38	α-Pinene	918	918	2.54 ± 0.12	1.85 ± 0.49	3.76 ± 0.07	3.72 ± 0.98	2.93 ± 0.65	3.77 ± 0.17	C_10_H_16_
3.	7.82	Camphene	933	933	0.03 ± 0.01	0.01 ± 0.02	0.06 ± 0.01	0.05 ± 0.01	0.09 ± 0.01	0.18 ± 0.01	C_10_H_16_
4.	8.61	β-Thujene	962	962	-	-	-	-	-	3.34 ± 0.10	C_10_H_16_
5.	8.68	β-Pinene	965	965	2.48 ± 0.11	1.80 ± 0.38	6.34 ± 0.06	3.53 ± 0.68	34.24 ± 3.50	19.62 ± 0.37	C_10_H_16_
6.	9.13	β-Myrcene	981	981	0.44 ± 0.02	0.16 ± 0.15	1.13 ± 0.07	1.10 ± 0.21	5.16 ± 0.21	0.89 ± 0.11	C_10_H_16_
7.	9.52	α-Phellandrene	995	995	0.06 ± 0.00	0.02 ± 0.03	0.13 ± 0.04	0.04 ± 0.00	0.20 ± 0.00	0.13 ± 0.09	C_10_H_16_
8.	9.72	3-Carene	1002	1002	0.06 ± 0.00	0.04 ± 0.01	0.03 ± 0.03	0.01 ± 0.01	0.02 ± 0.01	-	C_10_H_16_
9.	9.90	2-Carene	1008	1008	0.28 ± 0.01	0.20 ± 0.04	1.03 ± 0.04	0.45 ± 0.03	3.54 ± 0.16	0.69 ± 0.04	C_10_H_16_
10.	10.08	*p*-Cymene	1014	1014	2.45 ± 0.10	1.74 ± 0.31	2.29 ± 0.08	3.11 ± 0.02	-	2.29 ± 0.09	C_10_H_14_
11.	10.17	*m*-Mentha-6,8-diene	1017	1019	-	-	-	-	4.12 ± 0.18	-	C_10_H_16_
12.	10.29	_D_-Limonene	1020	1021	10.22 ± 0.18	7.85 ± 1.50	10.03 ± 0.27	12.49 ± 0.21	-	6.34 ± 0.04	C_10_H_16_
13.	10.59	trans-β-Ocimene	1030	1031	0.06 ± 0.01	0.10 ± 0.06	0.12 ± 0.01	0.41 ± 0.02	0.31 ± 0.01	0.06 ± 0.01	C_10_H_16_
14.	10.92	cis-β-Ocimene	1041	1041	0.36 ± 0.02	0.27 ± 0.02	8.29 ± 0.25	1.99 ± 0.01	11.31 ± 0.62	2.81 ± 0.10	C_10_H_16_
15.	11.23	γ-Terpinene	1051	1052	19.37 ± 0.3	21.04 ± 0.36	48.56 ± 1.01	19.42 ± 0.50	6.19 ± 0.31	17.92 ± 0.26	C_10_H_16_
16.	11.53	trans-Sabinenehydrate	1060	1062	-	-	-	-	0.22 ± 0.02	0.03 ± 0.01	C_10_H_18_O
17.	12.15	α-Terpinolene	1080	1082	0.61 ± 0.05	0.63 ± 0.03	3.31 ± 0.14	1.20 ± 0.14	1.12 ± 0.09	1.78 ± 0.07	C_10_H_16_
18.	12.28	*p*-Cymenene	1085	1085	-	-	-	-	-	0.41 ± 0.14	C_10_H_12_
19.	12.54	Linalool	1093	1093	-	-	5.18 ± 0.34	0.53 ± 0.06	16.8 ± 1.49	22.20 ± 0.43	C_10_H_18_O
20.	13.22	Cis-Sabinenehydrate	1115	1116	-	-	-	-	0.40 ± 0.05	0.02 ± 0.00	C_10_H_18_O
21.	14.85	Isocamphopinone	1167	1168	-	-	-	-	-	0.03 ± 0,00	C_10_H_16_O
22.	14.95	Terpinen-4-ol	1170	1171	0.24 ± 0.02	-	0.08 ± 0.13	0.19 ± 0.03	10.91 ± 1.35	0.81 ± 0.01	C_10_H_18_O
23.	15.38	α-Terpineol	1184	1184	-	-	-	0.17 ± 0.04	0.50 ± 0.05	0.79 ± 0.01	C_10_H_18_O
24.	16.64	Anisole	1227	1227	-	-	-	0.20 ± 0.02	-	1.11 ± 0.01	C_7_H_8_O
25.	18.38	Thymol	1288	1288	-	-	-	-	-	10.98 ± 0.27	C_10_H_14_O
26.	18.50	Carvacrol	1292	1292	-	-	-	0.36 ± 0.04	-	-	C_10_H_14_O
27.	19.59	σ-Elemene	1330	1330	-	-	-	0.01 ± 0.01	-	0.07 ± 0.01	C_15_H_24_
28.	21.12	β-Elemene	1384	1385	0.28 ± 0.01	0.24 ± 0.02	5.54 ± 0.10	0.11 ± 0.01	0.64 ± 0.06	0.06 ± 0.01	C_15_H_24_
29.	21.62	Dimethyl anthranilate	1401	1402	56.51 ± 1.06	58.77 ± 2.04	0.02 ± 0.02	49.06 ± 1.94	-	-	C_9_H_11_NO_2_
30.	21.90	β-Caryophyllene	1412	1413	2.90 ± 0.09	4.38 ± 0.51	1.36 ± 0.01	-	0.30 ± 0.03	0.82 ± 0.01	C_15_H_24_
31.	22.82	α-Caryophyllene	1448	1448	0.09 ± 0.01	0.19 ± 0.09	0.60 ± 0.01	0.19 ± 0.01	0.07 ± 0.01	0.09 ± 0.01	C_15_H_24_
32.	23.37	β-Chamigren	1470	1473	-	-	0.03 ± 0.00	-	-	-	C_15_H_24_
33.	23.53	Germacrene D	1476	1476	-	-	-	-	-	0.02 ± 0.00	C_15_H_24_
34.	23.69	β-Selinene	1482	1483	-	-	0.07 ± 0.01	-	-	-	C_15_H_24_
35.	23.94	δ-Guaiene	1492	1493	0.03 ± 0.00	0.04 ± 0.01	0.30 ± 0.01	0.08 ± 0.01	-	1.06 ± 0.02	C_15_H_24_
36.	24.18	α-Selinene	1501	1501	-	-	0.23 ± 0.01	-	-	-	C_15_H_24_
37.	24.26	α-Farnesene	1505	1505	-	-	-	-	-	0.22 ± 0.02	C_15_H_24_
38.	24.59	σ-Cadinene	1517	1517	-	-	0.04 ± 0.00	-	-	0.12 ± 0.00	C_15_H_24_
39.	26.18	Caryophyllene oxide	1579	1579	-	-	-	0.02 ± 0.00	-	0.02 ± 0.02	C_15_H_24_O
Monoterpene hydrocarbons (%)					39.93	36.38	86.45	49.07	69.96	61.31	
Oxygenated monoterpenes (%)					0.24	-	5.26	1.25	28.83	34.86	
Sesquiterpene hydrocarbons (%)					3.30	4.85	8.16	0.39	1.01	2.46	
Oxygenated sesquiterpenes (%)					-	-	-	0.02	-	0.02	
Others (%)					56.51	58.77	0.02	49.26	-	1.11	
Total identified (%)					99.98	100	99.89	99.99	99.80	99.76	

^a^ Kovats index determined experimentally on RTX-5 column relative to C8–C30 n-alkanes. ^b^ Published Kovats retention indices. Identification was based on comparison of the compounds mass spectral data (MS) and Kovats retention indices (RIs) with those of NIST Mass Spectral Library (2011), Wiley Registry of Mass Spectral Data 8th edition and literature.

**Table 3 plants-11-03335-t003:** Docking score of major compounds of *C. reticulata* cultivars with hyaluronidase, collagenase, and elastase binding sites.

Compound Name	Docking Score with Hyaluronidase (1fcv) Kcal/mole	Docking Score with Collagenase (456c) Kcal/mole	Docking Score with Elastase (6eoq) Kcal/mole
α-Pinene	−7.1	−6.8	−5.9
_D_-Limonene	−5.8	−7.3	−6.8
cis-β-Ocimene	−6.6	−7.3	−7.4
γ-Terpinene	−7.1	−7.1	−6.9
Linalool	−8.1	−7.9	−9.1
Terpinen-4-ol	−7.3	−7.4	−6.7
Thymol	−9.8	−10.2	−12.3
Dimethyl anthranilate	−8.2	−8.6	−8.0

## Data Availability

Data supporting the reported results can be found at, NIST Chemistry Webbook, https://webbook.nist.gov/chemistry/, (accessed on 10 September 2022).

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
