# Peer review of "Citrus reticulata Leaves Essential Oil as an Antiaging Agent: A Comparative Study between Different Cultivars and Correlation with Their Chemical Compositions"

_plants, 2022, doi:10.3390/plants11233335_

Round 1

Reviewer 1 Report

1. The subject addressed is one of interest, Mass-based metabolomic approach was implemented using GC-MS coupled with chemometric analysis to discriminate between the essential oil compositions of six cultivars of Citrus reticulata.

2. The subject is original because it addresses the antiaging capability of the essential oils were investigated through measurement of their ability to inhibit the major enzymes hyaluronidase, collagenase, and amylase involved in aging.

3. As a novelty for the addressed field: metabolic profiles of Citrus reticulata leaves cultivar EO by GC-MS, chemometric Analysis Based on GC-MS Analysis, HCA dendogram (c) based on GC-MS metabolic profiles of different Citrus cultivars based on the identification of volatile compounds.

4. As a suggestion for the authors, I think that the analysis of phenolic compounds, the antioxidant capacity can bring a plus to this research

5. I recommend introducing the conclusion in the manuscript, it is missing

6. References are adequate, but I suggest using newer references

Author Response

Response to Reviewer 1 Comment

Comment 1: The subject addressed is one of interest, Mass-based metabolomic approach was implemented using GC-MS coupled with chemometric analysis to discriminate between the essential oil compositions of six cultivars of Citrus reticulata.

Response: We would like to thank the reviewer for his comment.

Comment 2: The subject is original because it addresses the antiaging capability of the essential oils were investigated through measurement of their ability to inhibit the major enzymes hyaluronidase, collagenase, and amylase involved in aging.

Response: We would like to thank the reviewer for his comment.

Comment 3: As a novelty for the addressed field: metabolic profiles of Citrus reticulata leaves cultivar EO by GC-MS, chemometric Analysis Based on GC-MS Analysis, HCA dendogram (c) based on GC-MS metabolic profiles of different Citrus cultivars based on the identification of volatile compounds.

Response: We would like to thank the reviewer for his comment.

Comment 4: As a suggestion for the authors, I think that the analysis of phenolic compounds, the antioxidant capacity can bring a plus to this research

Response: The authors highly appreciate the reviewer comment. We would like to clarify that essential oils as a class of active constituents are characterized by hydrocarbons and oxygenated hydrocarbons nucleus which are best analyzed using GC-MS. Concerning the antioxidant activity, we build our study supported by the fact that citrus species extracts and essential oils are well-known for their antioxidant capacity [1-3], and thus proving their anti-collagenase, anti-hyaluronidase, and anti-elastase activities will make them a promising antiaging candidate.  This was clarified in the revised manuscript in the introduction section P.2, Line: 63-65

Comment 5: I recommend introducing the conclusion in the manuscript, it is missing

Response: We would like to thank the reviewer for his valuable suggestion. A conclusion part was added P. 12, Line: 376-385

Comment 6: References are adequate, but I suggest using newer references

Response: We would like to thank the reviewers for his comment. Recent references were added. P. 1 Line 44, P. 2, Line 45

References

  1. Denkova-Kostova, R.; Teneva, D.; Tomova, T.; Goranov, B.; Denkova, Z.; Shopska, V.; Slavchev, A.; Hristova-Ivanova, Y. J. Z. f. N. C. Chemical composition, antioxidant and antimicrobial activity of essential oils from tangerine (Citrus reticulata L.), grapefruit (Citrus paradisi L.), lemon (Citrus lemon L.) and cinnamon (Cinnamomum zeylanicum Blume). Zeitschrift für Naturforschung C 2021 76, (5-6), 175-185.
  2. Dalia, I. H.; Maged, E. M.; Assem, M. E.-S. J. J. o. M. P. R. Citrus reticulata Blanco cv. Santra leaf and fruit peel: A common waste products, volatile oils composition and biological activities. Journal of Medicinal Plants Research 2016 10, (30), 457-467.
  3. Chi, P. T. L.; Van Hung, P.; Le Thanh, H.; Phi, N. T. L. J. W.; Valorization, B. Valorization of citrus leaves: Chemical composition, antioxidant and antibacterial activities of essential oils. Waste and Biomass Valorization 2020 11, (9), 4849-4857.

Reviewer 2 Report

Manuscript Number: plants-2050647, titled:

Citrus reticulata Leaves Essential Oil As An Antiaging Agent: A Comparative Study between Different Cultivars and Correlation with Their Chemical Compositions

Review 1 – 18 November 2022

Dear Editor of Plants

1)     the argument is interesting but it has to be improved. The introduction section has to be improved and better detailed with proper references. The M&M section has to be completed with the agronomic factors of cultivation. The references section is not arranged as per Plants instructions for authors. Many inaccuracies in the text;

I suggest a major revision

To the Authors (in detail):

2)     the argument is interesting but it has to be improved. The introduction section has to be improved and better detailed with proper references. The M&M section has to be completed with the agronomic factors of cultivation. The references section is not arranged as per Plants instructions for authors. Many inaccuracies in the text;

3)     Introduction section, after lines 31-33, please, explain that the essential oil composition from different parts of the citrus plants is related with many factors such as harvest year [X1]; harvest date [X2]; cultivar [X3]; rootstocks [X4]; extraction system [X5-X6]. Please, find, read and discuss the following studies and include the reference number after each statement and not at the end of the sentence:

[X1] The peel essential oil composition of bergamot fruit (Citrus bergamia, Risso) of Reggio Calabria  (Italy): a review.

Emirates Journal of Food and Agriculture  32 (11) 835-845 (2020)

doi: 10.9755/ejfa.2020.v32.i11.2197

[X2] Composition and seasonal variation of the essential oil from leaves and peel of a Cretan lemon variety.

J Agric Food Chem.2002 Jan 2;50(1):147-53.

doi: 10.1021/jf001369a.

[X3] Citrus bergamia, Risso: the peel, the juice and the seed oil of the bergamot fruit of Reggio Calabria (South Italy).

Emirates Journal of Food and Agriculture 32(7) 522-532 (2020).

DOI: 10.9755/ejfa.2020.v32.i7.2128

[X4] Essential oil components of Citrus cultivar ‘MALTAISE DEMI SANGUINE’ (Citrus sinensis) as affected by the effects of rootstocks and viroid infection

International Journal of Food Properties

2019, VOL. 22, NO. 1, 438–448

https://doi.org/10.1080/10942912.2019.1588296

[X5] Effect of distillation methods on the leaf essential oil of some Citrus cultivars.

Journal of Essential Oil Research 33(5):452-463 (2021).

DOI: 10.1080/10412905.2021.1936666

[X6] Cold Pressing, Hydrodistillation and Microwave Dry Distillation of Citrus Essential Oil from Algeria: A Comparative Study.

 Electronic Journal of Biology, 2016, Vol.S1: 30-41

4)     Introduction section, lines 44-45, please, explain the Country of origin, related with Citrus reticulata;

5)     Introduction section, line 46 (varieties), line 60 (cultivars): Variety and cultivar are not synonyms, please, in the whole manuscript, specify if you are talking about cultivar or variety;

6)     Line 76: Kovat?

7)     Lines 99, 108, 113 and in the whole manuscript, when you insert a reference, please, use the instructions for authors of Plants;

8)     Line 138: (1a and 1b);

9)     Line 143, please, specify, Citrus reticulata cultivars;

10) Line 165 (2a and 2b);

11) Line 178: Am, Bm and Wm. In the whole manuscript, replace “&” with “and”;

12) Figure 1b: γ-Terpinene and not γ-Terpinen;

13) Figure 2b: γ-Terpinene and not γ-Terpinen;

14) Sub-section 3.1, please, include data related with: type of soil; microclimate (rain, temperatures); age of plants; number of plants per each cultivar used for leaves sampling; distance of plants on the line and between the lines; fertilizers used (type, quantity, period); irrigation (type, quantity, period);

15) Sub-section 3.1, include the geographical coordinates of the cultivation area and harvest year;

16) Sub-section 3.1, please specify the ripening index of fruits or days after blossom, or °Brix of fruits, when leaves were sampled;

17) Sub-section 3.2, did you used the entire leave or have you cut it in small pieces?

18) Line 296, when you indicate the column parameters, include the word length after 30 m, for consistency with other parameters;

19) Sub-section 3.2, please, report the type of stationary phase of the GC column;

20) Line 301 and in the whole manuscript, when you indicate a temperature, separate the numeric value by the symbol: 280 °C and not 280°C

21) Line 303, verify the spacing between words and punctuation, before… Different;

22) Sub-section 3.2, describe how long between sampling of leaves and hydro-distillation; describe also how long between essential oil extraction and GC-MS analysis;

23) Line 223 (ml), line 328 (μL), line 337(μl) decide if to use L in capital or small letter and be consistent in the whole manuscript;

24) References section: the references are not reported as per the instructions for authors of Plants, for example: the journal name has to be abbreviated with the official abbreviation, the issue number is not required, and so on;

25) References number 14 and in the whole section, the species in small letter: reticulata and not Reticulata;

26) Please, write in blue color or evidence differently the corrections you will do.

I suggest a major revision

Regards.

Author Response

Response to Reviewer 2 Comments

Comment 1: the argument is interesting but it has to be improved. The introduction section has to be improved and better detailed with proper references. The M&M section has to be completed with the agronomic factors of cultivation. The references section is not arranged as per Plants instructions for authors. Many inaccuracies in the text;

Response: We are grateful for the reviewer valuable comments and suggestions. All comments were addressed point-by-point in the revised manuscript.

Comment 2: Introduction section, after lines 31-33, please, explain that the essential oil composition from different parts of the citrus plants is related with many factors such as harvest year [X1]; harvest date [X2]; cultivar [X3]; rootstocks [X4]; extraction system [X5-X6]. Please, find, read and discuss the following studies and include the reference number after each statement and not at the end of the sentence:

[X1] The peel essential oil composition of bergamot fruit (Citrus bergamia, Risso) of Reggio Calabria  (Italy): a review. Emirates Journal of Food and Agriculture  32 (11) 835-845 (2020)

doi: 10.9755/ejfa.2020.v32.i11.2197

[X2] Composition and seasonal variation of the essential oil from leaves and peel of a Cretan lemon variety.

J Agric Food Chem.2002 Jan 2;50(1):147-53.

doi: 10.1021/jf001369a.

[X3] Citrus bergamia, Risso: the peel, the juice and the seed oil of the bergamot fruit of Reggio Calabria (South Italy).

Emirates Journal of Food and Agriculture 32(7) 522-532 (2020).

DOI: 10.9755/ejfa.2020.v32.i7.2128

[X4] Essential oil components of Citrus cultivar ‘MALTAISE DEMI SANGUINE’ (Citrus sinensis) as affected by the effects of rootstocks and viroid infection

International Journal of Food Properties

2019, VOL. 22, NO. 1, 438–448

https://doi.org/10.1080/10942912.2019.1588296

[X5] Effect of distillation methods on the leaf essential oil of some Citrus cultivars.

Journal of Essential Oil Research 33(5):452-463 (2021).

DOI: 10.1080/10412905.2021.1936666

[X6] Cold Pressing, Hydrodistillation and Microwave Dry Distillation of Citrus Essential Oil from Algeria: A Comparative Study.

 Electronic Journal of Biology, 2016, Vol.S1: 30-41

Response: The authors highly appreciate the reviewer comment, all mentioned referenced were carefully read and added in the revised manuscript. P.1 line 36-39.

Comment 3. Introduction section, lines 44-45, please, explain the Country of origin, related with Citrus reticulata.

Response: We would like to thank the reviewers for his comment. It was explained in the revised manuscript P.2, line: 48-50.

Comment 4: Introduction section, line 46 (varieties), line 60 (cultivars): Variety and cultivar are not synonyms, please, in the whole manuscript, specify if you are talking about cultivar or variety;

Response: We would like to thank the reviewer for his/her remark. It was corrected to cultivar in the revised manuscript.

Comment 5. Line 76: Kovat

Response: We would like to clarify that Kovat index is used to converts retention times into system independent constants thus, allows comparing values measured by different analytical laboratories under varying conditions using homologous series of n-alkanes as standards against which adjusted retention times are measured for compounds of interest.

Comment 6: Lines 99, 108, 113 and in the whole manuscript, when you insert a reference, please, use the instructions for authors of Plants;

Response: We would like to clarify that those sentences we are mentioning the authors who reported this result and not using it as a citation, as the citation is inserted at the end of those sentence numbered as Plants style.

Comment 7: Line 138: (1a and 1b);

Response: Done.

Comment 8: Line 143, please, specify, Citrus reticulata cultivars;

Response: Specified.

Comment 9: Line 165 (2a and 2b);

Response: Done

Comment 10: Line 178: Am, Bm and Wm. In the whole manuscript, replace “&” with “and”;

Response: Done.

Comment 11: Figure 1b: γ-Terpinene and not γ-Terpinen;

Response: Corrected

Comment 12: Figure 2b: γ-Terpinene and not γ-Terpinen;

Response: corrected

Comment 13: Sub-section 3.1, please, include data related with: type of soil; microclimate (rain, temperatures); age of plants; number of plants per each cultivar used for leaves sampling; distance of plants on the line and between the lines; fertilizers used (type, quantity, period); irrigation (type, quantity, period);

Response: The type of the soil is light clay soil, (Slight rain at 25 °C). The trees are 20 years old, where, approximately from 5 to 7 trees were used for leaves sampling for each cultivar. Distance of plants between the lines were approximately 1.5 m and 1 m on the line.

Phosphate and organic fertilizers and agricultural sulfur are used. Phosphate fertilizers are added in the form of mono-superphosphate at a rate of 30 kg per feddan, during the months of December and January mixed with fully decomposed municipal fertilizers (15-20 m3 per feddan) and 100 kg of agricultural sulfur until it decomposes before the spring season. Trees are irrigated by immersion, 3000-5000 cm3from 30-45 days.

This data was provided as a supplementary file as it is not relevant to the context.

Comment 14: Sub-section 3.1, include the geographical coordinates of the cultivation area and harvest year;

Response: Geographical coordinates and harvest year were included in the revised manuscript P.  10, line 291-295

Comment 15: Sub-section 3.1, please specify the ripening index of fruits or days after blossom, or °Brix of fruits, when leaves were sampled;

Response: This was clarified in the revised manuscript P.10, line 291

Comment 16: Sub-section 3.2, did you used the entire leave or have you cut it in small pieces?

Response: Entire leaves were used, clarified in the revised manuscript

Column 17: Line 296, when you indicate the column parameters, include the word length after 30 m, for consistency with other parameters;

Response: Done.

Comment 18: Sub-section 3.2, please, report the type of stationary phase of the GC column;

Response: It was mentioned

Comment 19: Line 301 and in the whole manuscript, when you indicate a temperature, separate the numeric value by the symbol: 280 °C and not 280°C

Response: We would like to thank the reviewer for this remark, it is corrected.

Comment 20: Line 303, verify the spacing between words and punctuation, before… Different.

Response: We would like to thank the reviewer for this remark, it is corrected.

Comment 21: Sub-section 3.2, describe how long between sampling of leaves and hydro-distillation; describe also how long between essential oil extraction and GC-MS analysis;

Response: Described in the revised manuscript.

Comment 22: Line 223 (ml), line 328 (μL), line 337(μl) decide if to use L in capital or small letter and be consistent in the whole manuscript;

Response: Corrected in the revised manuscript.

Comment 23: References section: the references are not reported as per the instructions for authors of Plants, for example: the journal name has to be abbreviated with the official abbreviation, the issue number is not required, and so on;

Response: All references were checked and updated following Plants style.

Comment 24: References number 14 and in the whole section, the species in small letter: reticulata and not Reticulata;

Response: Done

Comment 25: Please, write in blue color or evidence differently the corrections you will do.

Response: All changes are highlighted in yellow.

Round 2

Reviewer 2 Report

Manuscript Number: plants-2050647, titled:

Citrus reticulata Leaves Essential Oil As An Antiaging Agent: A Comparative Study between Different Cultivars and Correlation with Their Chemical Compositions

Review 2 – 25 November 2022

Dear Editor of Plants

the authors have included many of my comments, anyway, some improvement more is necessary.

I suggest a minor revision

To the Authors (in detail):

1)    you have included many of my comments, anyway, some improvement more is necessary;

2)    Introduction section, line 36: worth noting and not woth noting;

3)    Lines 83-84: Kovats indices and not Kovat indices;

4)    Sub-section 3.1, please move the information about the agronomic conditions from supplementary material to Material and Methods section. These are very important because the essential oil composition is also related with the agronomic conditions. The reader wants to immediately read all information. Agronomic conditions and not agronomy conditions. Describe also the microclimatic condition during the year: temperature and rain during the year;

5)    What means: “Trees are irrigated by immersion, 3000-5000 cm3 from 30-45 days? Please, explain well;

6)    Sub-section 3.1, please, indicate when the sampling was conducted: days after flowering and the week of the year;

7)    Please, write in blue color or evidence differently the corrections you will do.

I suggest a minor revision

Regards.

Author Response

Response to Reviewer 2 Comments

  • you have included many of my comments, anyway, some improvement more is necessary;

Response: The authors highly appreciate the reviewer comment, all comments are considered

  • Introduction section, line 36: worth noting and not woth noting;

Response: Corrected

  • Lines 83-84: Kovats indices and not Kovat indices;

Response: Corrected

  • Sub-section 3.1, please move the information about the agronomic conditions from supplementary material to Material and Methods section. These are very important because the essential oil composition is also related with the agronomic conditions. The reader wants to immediately read all information. Agronomic conditions and not agronomy conditions. Describe also the microclimatic condition during the year: temperature and rain during the year;

Response: all information is added to the manuscript

  • What means: “Trees are irrigated by immersion, 3000-5000 cm3from 30-45 days? Please, explain well; 

Response: explained

  • Sub-section 3.1, please, indicate when the sampling was conducted: days after flowering and the week of the year;

Response: 10 days after flowering (10th week of the year)

  • Please, write in blue color or evidence differently the corrections you will do.

Response: all changes are highlighted in green.
